# Physicochemical Characterization of an Exopolysaccharide Produced by *Lipomyces* sp. and Investigation of Rheological and Interfacial Behavior

**DOI:** 10.3390/gels7040156

**Published:** 2021-09-28

**Authors:** Wentian Li, Yilin Guo, Haiming Chen, Wenxue Chen, Hailing Zhang, Ming Zhang, Qiuping Zhong, Weijun Chen

**Affiliations:** 1College of Food Sciences & Engineering, Hainan University, Haikou 570228, China; 19085231210019@hainanu.edu.cn (W.L.); 20190881310118@hainanu.edu.cn (Y.G.); chwx@hainanu.edu.cn (W.C.); 995065@hainanu.edu.cn (M.Z.); 990511@hainanu.edu.cn (Q.Z.); 2Maritime Academy, Hainan Vocational University of Science and Technology, Haikou 571126, China; 3College of Life Sciences, Yantai University, Yantai 264005, China; hlzhang@ytu.edu.cn; 4Chunguang Agro-Product Processing Institute, Wenchang 571333, China

**Keywords:** *Lipomyces starkeyi*, exopolysaccharide, rheological property, interfacial property

## Abstract

The present study aimed to evaluate the rheological and interfacial behaviors of a novel microbial exopolysaccharide fermented by *L. starkeyi* (LSEP). The structure of LSEP was measured by LC-MS, ^1^H and ^13^C NMR spectra, and FT-IR. Results showed that the monosaccharide composition of LSEP was D-mannose (8.53%), D-glucose (79.25%), D-galactose (7.15%), and L-arabinose (5.07%); there existed the anomeric proton of α-configuration and the anomeric carbon of α- and β-configuration; there appeared the characteristic absorption peak of the phosphate ester bond. The molecular weight of LSEP was 401.8 kDa. The water holding capacity (WHC, 2.10 g/g) and oil holding capacity (OHC, 12.89 g/g) were also evaluated. The results of rheological properties showed that the aqueous solution of LSEP was a non-Newtonian fluid, exhibiting the shear-thinning characteristics. The adsorption of LSEP can reduce the interfacial tension (11.64 mN/m) well and form an elastic interface layer at the MCT–water interface. Such functional properties make LSEP a good candidate for use as thickener, gelling agent, and emulsifier to form long-term emulsions for food, pharmaceutical, and cosmetic products.

## 1. Introduction

Tender coconut water, which contains multiple water-soluble vitamins (vitamin C, B_1_, B_2_, B_3_, B_5_, B_6_, B_7_, and B_9_) and multiple inorganic ions (calcium, iron, magnesium, potassium, sodium, manganese, zinc, copper, phosphorous), is a refreshing natural drink and is deeply enjoyed by consumers [1]. Meanwhile, matured coconut water is not considered edible directly and is usually used as natural medium to produce nata de coco by fermentation (*Acetobacter xylinus*) at room temperature (30–37 °C) [2,3]. However, in some areas (such as Hainan Island), additional energy is needed for the growth and fermentation of *Acetobacter xylinus*, leading to the higher production cost of nata de coco. Therefore, in most cases, matured coconut water is always wasted as a by-product, causing serious waste of resources and environmental pollution [4].

The yeast, *L. starkeyi*, grows relatively fast with no seasonal limitation and is suitable for growing at room temperature [5]. *L. starkeyi* can utilize carbohydrates in wastewater and achieve a high cell density to efficiently produce microbial lipids under fed-batch culture conditions [6]. Although *L. starkeyi* were mainly used for lipid production, some strains of them can also produce exopolysaccharides under suitable conditions. Thirumal et al. [7] reported on a strain of *L. starkeyi* with a focus on the effects of a low-cost medium on lipid and exopolysaccharide production under repeated fed-batch and continuous cultivation. Even so, almost all investigations of *L. starkeyi* still focused on the production of lipids.

The microbial polysaccharides are special polymers with excellent biological and physicochemical properties, so they are often used as thickener, gelling agent, emulsifier, stabilizer suspending agent, and drug carrier [8,9,10,11]. The rheological behavior of exopolysaccharides showed that many microbial exopolysaccharides exhibited good gelation and could be used as potential gelling agents in industrial production, for example, the exopolysaccharides produced by *Mesorhizobium loti* Semia 816, *Sphingomonas sanxanigenens* NX02, and *Pseudomonas stutzeri* AS22 [9,10,11]. As a gelling agent, microbial exopolysaccharides have the advantages of wide source, low production cost, good biocompatibility, and biodegradability. Although research on microbial exopolysaccharides is extensive, the fermentation of matured coconut water to produce exopolysaccharides is still unexplored. Moreover, there is still no research on the structures and properties of exopolysaccharides produced by *L. starkeyi*.

The major objective of this study was to investigate the rheological and interfacial behaviors of a novel microbial exopolysaccharide fermented by *L. starkeyi* (LSEP) in the matured coconut water medium. In addition, the structure and monosaccharide composition of LSEP were also determined to evaluate the structure–function relationship of LSEP. LSEP as a potential gelling agent can improve the viscosity of the system, keep the system in a uniform and stable suspended (or opacified) state, or form a gel. The results suggest that LSEP would be a strong stabilizer or emulsifying agent.

## 2. Results and Discussion

### 2.1. LSEP Preparation

LSEP was obtained by *L. starkeyi* fermentation in matured coconut water. As shown in Figure 1a, the cell grew rapidly in the first 24 h, indicating that the cell was still in logarithmic phase. There was no significant change of the OD_570nm_, which reflected the cell density in the last 6 days, indicating that the cell was in a stationary phase. Such a two-stage growth and fermentation process coincided with previous related research [7,12]. The residual glucose and the accumulation of LSEP during fermentation are also shown in Figure 1b,c, respectively. It is worth noting that the initial glucose concentration was 111.68 g/L instead of 90 g/L, because the mature coconut water itself contains a certain amount of glucose. As shown in Figure 1c, the accumulation of LSEP reached maximum (2.85 g/L) after fermentation for 6 days. Because the cell began to accumulate and produce LSEP during the cultivation of the seed, the fermentation medium already contained a certain amount of LSEP at the beginning. The cells (logarithmic phase) multiplied and grew rapidly, resulting in a lot of carbon source consumption. According to the report of Thirumal et al. [7], the decrease of LSEP is due to the depletion of the cells themselves. Therefore, the content of LSEP decreased at the initial stage of fermentation.

The crude LSEP was first purified using a DEAE–Sepharose fast flow column (Figure 2a). Figure 2a shows that the main peak fraction appeared when the concentration of NaCl was 0, indicating that the principal component of crude LSEP was a kind of a neutral polysaccharide. The desired peak fraction was collected and set for further purification by size-exclusion chromatography Sephadex G-200 (Figure 2b). The fraction corresponding to the desired peak was collected and lyophilized to get purified LSEP.

### 2.2. Monosaccharide Composition

The result of the monosaccharide composition of LSEP is shown in Table 1. The monosaccharides of LSEP were D-mannose (8.53%), D-glucose (79.25%), D-galactose (7.15%), and L-arabinose (5.07%). According to the peak area and the molecular weight of each monosaccharide, the molar ratio of the four monosaccharides was calculated to be 1.4:13.1:1.2:1. Generally speaking, the monosaccharide compositions of many microbial exopolysaccharides contain glucose, mannose, and galactose [13,14,15,16]. Thirumal et al. [7] demonstrated that the exopolysaccharide produced by *L. starkeyi* may contain mannose and galactose. Compared with the study of Mangala et al. [17], it could be found that although exopolysaccharides were both yielded by *L. starkeyi*, their monosaccharide compositions were different. These differences were probably caused by genetic differences between different strains, and the different compositions of monosaccharides will lead to different structural and functional properties.

### 2.3. Molecular Weight Analysis

The GPC analysis was used to estimate the molecular weight and homogenization of the LSEP. The GPC curve (Figure 3a) showed a single and narrow peak, indicating that the purified LSEP was a homogeneous polysaccharide with high purity. By calculating the standard curve made by the standard dextran, the molecular weight of the LSEP was about 400 kDa. Studies illustrated that the activity of exopolysaccharides was closely related to molecular weight. Low-molecular-weight exopolysaccharides often have excellent antioxidant, antibacterial, and even anticancer activities [18], while high-molecular-weight exopolysaccharides often have higher viscosity because of their complex structures [9,19]. Considering the LSEP (400 KDa) belonged to a high-molecular-weight microbial exopolysaccharide, it might have good rheological properties.

### 2.4. FT-IR Analysis

FT-IR spectroscopy has been proved to be the most important analytical method to identify functional groups, which can infer some possible characteristic structures in the polysaccharide component based on the characteristic absorption peaks [20]. As shown in Figure 3b, the broad and large absorption peak at 3405.91 cm^−1^ was caused by the stretching vibration of −OH, which was the characteristic absorption peak of polysaccharides [7]. This wide bands correspond to the hydroxyl groups with intermolecular hydrogen bonding, which mean that there are a lot of intermolecular hydrogen bonds in the LSEP, thus affecting the viscosity of the polysaccharide solution. The absorption peak at 2931.08 cm^−1^ was caused by the asymmetric stretching vibration of the C–H bond in the alkyl group of the polysaccharide chain structure [21]. The band at 1639.63 cm^−1^ was attributed to the asymmetric stretching vibration of the C=O bands [22]. The 1500–1200 cm^−1^ region was assigned to the O–H/C–H deformation and C–H/C–H_2_ stretching vibrations [23]. The peaks at 1200–1000 cm^−1^ were attributed to C–O–C and C–O–H linkages [24]. The strong and sharp absorption peak at 1017.03 cm^−1^ came from the stretching vibration of C–O, which was also a characteristic absorption peak of polysaccharides [25]. The absorption peak at 547.49 cm^−1^ was the characteristic absorption peak of the phosphate ester bond [26]. Considering a large amount of PO^−4^ was contained in the fermentation medium, it was reasonable that the absorption peak of the phosphate ester bond appeared in the FT-IR of LSEP.

### 2.5. NMR Analysis

One-dimensional NMR (^1^H and ^13^C NMR) is commonly used to recognize the α- or β-configuration of the anomeric proton (H1) and anomeric carbon (C1) in sugar residue [27,28,29]. In most cases, the anomeric region of α-configuration appears in 5.1–5.8 ppm (anomeric proton) and β-configuration relatively corresponds to 4.3–4.8 ppm [27]. The 4.50–3.00 ppm region was correlated to the ring proton region [30]. It can be seen from the ^1^H NMR spectrum of LSEP (Figure 3c) that the signal at 5.30 ppm represented the anomeric proton of α-configuration. The signals in the range of 4.20–3.40 ppm indicated the existence of numerous ring protons. The complexity and heterogeneity of LSEP were confirmed by the presence of high-field chemical shift signals near 4.08–3.37 ppm, indicating that LSEP may have complex intermolecular interactions, which may affect the viscosity and viscoelasticity of the LSEP solution [31].

The anomeric resonances for the anomeric carbons appear at δ 90 to 110 ppm, whereas those for the ring carbons are found between 60 and 85 ppm. Specifically, the chemical shifts between 95 and 101 ppm were due to α-linked residue and the chemical shifts in the region of 101–105 ppm were attributed to the β-linked residue [32]. As shown in Figure 3d, the signals at 97.81 and 104.27 ppm were attributed to the anomeric region of the α- and β-configuration, respectively. The signals at the range of 57–87 ppm indicated the existence of C2–C6 [27].

### 2.6. X-ray Diffraction Analysis

Most polysaccharides have an amorphous or semi-crystalline structure. Since intermolecular bonds are weaker in amorphous regions rather than in crystalline regions, the solubility and water uptake are higher in the amorphous polysaccharides [32]. It can be seen from the Figure 3e that there were five diffraction peaks (27.36, 31.58, 45.31, 56.33, 66.17) and the corresponding crystal plane spacings were 3.26, 2.83, 2.00, 1.63, and 1.41 Å. The crystallinity of LSEP was 60.48%, and it showed a semi-crystalline structure. That semi-crystalline structure depended on the degree of order within the material, which may affect certain physical properties of polysaccharides, such as tensile strength, flexibility, solubility, and swelling power [33]. Therefore, the semi-crystalline structure of LSEP may have a distinctive effect on its viscosity and viscoelasticity in aqueous solution.

### 2.7. Thermal Analysis

The thermal properties of polysaccharides are of great significance for evaluating their applications in industry and food processing [34,35]. Figure 3f shows the TG and DTG curves of LSEP in the range of 30–650 °C, and the weight loss of polysaccharides occurred in two stages. According to the DTG curve, there were four valleys at 31.2, 65.5, 253.5 and 301.6 °C, respectively. In the range of 31.2–65.5 °C was the first stage of weight loss, and the second stage of weight loss occurred at 253.5–301.6 °C. The weight loss of the two stages was 5.34% and 25.16%, respectively. The weight loss of LSEP in the first stage mainly came from the loss of water, combining with the hydroxyl group in the polysaccharide structure, and the weight loss in the second stage was mainly caused by the depolymerization of the polysaccharide chain structure, which was the breakage of the C–C bond and the C–O bond. The largest weight loss of LSEP occurred in the second stage, which was related to the oxidation of residual organic matter after the first stage of pyrolysis [36,37]. The stable temperature of LSEP was 253.5 °C, which was higher than that of polysaccharide from *Ocimum album* seed, *Leu. lactis* KC117496, and *Halomonas* sp. AAD6 [32,38,39]. The high stable temperature of LSEP meant it could be used in food processing systems, wherein temperatures seldom exceed 150 °C [40]. Further, the good thermal stability of LSEP allows it to maintain its excellent rheological properties during higher-temperature processing.

### 2.8. Scanning Electron Microscopy Analysis

Scanning electron microscopy (SEM) analysis is helpful to study the surface morphology of polysaccharides to help understand their physical properties and explain their physicochemical properties [41]. The scanning electron micrograph of LSEP at 10,000× is shown in Figure 3g. The chain skeleton of LESP was a loose and porous, cross-linked network structure. The constituent units were granular and attached to the network structure. Such a cross-linked network structure will expose a large number of hydroxyl groups in the polysaccharide structure, indicating a good water holding capacity [42]. In addition, the porous structure will increase the viscosity of polysaccharides, which will make them have potential applications as food thickeners to improve the physical properties of products [43].

### 2.9. Water Holding Capacity (WHC) and Oil Holding Capacity (OHC)

WHC is an important functional indicator, which can reflect the amount of water absorbed and held by the hydrated samples [44]. As shown in Figure 3h, the WHC of LSEP was 2.10 g/g. As another functional indicator, the OHC of LSEP (12.89 g oil/g) was higher than those polysaccharides from *Lilium lancifolium* Thunb. (9.25 g oil/g) [45] and *Parkia speciosa* pod (3.9 g oil/g) [46]. The high OHC of LSEP was requisite for the formulated foods, because it was the critical determinant of flavor preservation.

### 2.10. Rheological Properties

Generally speaking, the investigation of the flow behavior of polysaccharides at low shear rate described the consistency of the foods in the mouth and illustrated the flow behavior during spraying, drying, or pumping fluids [47]. The results of rotary shear measurements (Figure 4a) showed the change of apparent viscosity with shear rate (0.1–100 s^−1^) at 25 °C. The LSEP solutions showed obvious shear-thinning at all concentrations (2%, 4%, 6%, 8%, and 10%), which may be attributed to the deformation of the chain structure, the reduction of the intermolecular interaction force, or even the reduction of the Mw during shearing [9]. As shown in Figure 4b, LSEP aqueous solution showed significant pseudoplastic characteristics and typical non-Newtonian fluid properties. This is similar to the shear-thinning phenomenon and pseudoplastic fluid behavior of other microbial polysaccharide aqueous solutions [11,48,49]. A shear-thinning phenomenon of water is commonly seen in a rotational rheometer because of the surface tension between the rotor and the plate. Therefore, it will affect the shear-thinning characteristics of samples in practical measurements. While, considering that LSEP is a surfactant (this can be proved by Section 2.11.1), it can significantly decrease the surface tension of water. Therefore, there was little influence on the sample under test. Since shear-thinning fluids are easily pumped and high pseudoplasticity produces a smooth mouthfeel, the LSEP is extensively used in food and pharmaceutical processing [47].

The elastic modulus (G′) and viscous modulus (G″) versus angular frequency (ω) are shown in Figure 4c,d. Commonly, the intercepts (K′ and K″) and magnitudes of the slopes (n′ and n”) are used to describe viscoelastic characteristics of food materials. Therefore, the slope of a double-logarithmic plot of modulus as a function of angular frequency was analyzed. As shown in Table 2, K′ and K″ increased with the increase of concentration. Furthermore, the positive slopes (n′ = 0.39933–0.55179 and n” = 0.32121–0.49331) of G′ and G″ confirmed the weak gel-like behavior of LSEP solutions. Furthermore, both n′ and n″ were close to zero, suggesting that G′ and G″ exhibited a slight frequency dependence. That is to say, the viscoelasticity of the samples depended on the relative domination of G′ and G″. Considering that G′ was higher than G″ in the individual linear viscoelastic region (LVR) (Appendix A), a certain elastic behavior (gel-like) was observed. Specifically, the difference between G′ and G″ expanded with the increase of concentration, suggesting that the gel performance of the LSEP solution became stronger as the concentration increased [50]. Due to the above rheological properties, LSEP has great potential in the food industry [51].

### 2.11. Interfacial Adsorption and Dilatational Rheological Properties

#### 2.11.1. Adsorption Kinetics and Molecular Rearrangements at the MCT–Water Interface

In general, the dynamic adsorption of colloidal particles at the oil–water interface mainly involves diffusion, penetration, and reorganization [52]. In pure biopolymer systems, the adsorption kinetics can be deduced from the π–t^1/2^ curves, and the slopes of these plots are the diffusion rate constant (K_diff_) [53]. The square root of time (t^1/2^) dependence of interfacial pressure (π) at the MCT–water interface is shown in Figure 5a. The π value increased with the adsorption time, which suggested that the amounts of LSEP absorbed at the MCT–water interface gradually increased [54]. For the adsorption kinetics of LSEP at the MCT–water interface, the K_diff_, K_p_, and K_r_ are shown in Table 3. There was no significant difference in the adsorption behavior, which indicated that the change of concentration had no significant effect on the K_diff_. Meanwhile, K_P_ and K_r_ showed different trends from the K_diff_, illustrating that the concentration mainly affected the penetration and reorganization. The highest K_P_ and K_r_ of LSEP occurred at 4%, demonstrating the strong molecular interaction, penetration, and reorganization.

#### 2.11.2. Dilatational Rheological Properties and Lissajous Plots

The viscoelastic properties reflect the adsorption of molecules at the interface and the interaction between molecules [55]. The evolution of dilatational viscoelasticity modulus (E) with time and π at MCT-water interface was shown in Figure 6. In all cases, E increased rapidly and then reached a relative equilibrium value. The E value at a concentration of 4% was the highest among all concentrations, indicating that it exhibits higher interactions with the adsorbed molecules [56]. Additionally, the slope of the E-π curve was used to evaluate the degree of molecular interactions at the interface [57]. The slope values were higher than 1.0, suggesting that nonideal adsorption behavior occurred at the oil-water interface [58]. The slope of the E-π curve was the closest to 1.0 at 6% concentration, indicating a possible ideal adsorption behavior. The highest slope (4%) indicated the strongest molecular interaction, which was consistent with the evolution of E with time.

Lissajous plots were used to investigate the nonlinear interfacial rheology under large deformation, which could help understand the relationship between interfacial properties and surface microstructure [59,60,61,62]. The Lissajous plots of surface pressure against deformation during amplitude sweeps for LSEP at different concentrations are shown in Figure 7. It was worth noting that the interfacial pressure (Π, defined as the surface tension of the deformed interface subtracts the surface tension of the nondeformed interface) at a concentration of 0.4% was always the lowest at the same amplitude, indicating that the interface was the most stable at this concentration. The Lissajous plots of all concentrations showed obvious asymmetries as the amplitude increased, which indicated that the response under compression and expansion was different. The Lissajous plots (0.6%, 0.8%, and 1.0%) narrowed obviously in the direction of maximum compression, which indicated that the interface layer tended to respond to a strain hardening during compression. In addition, the slopes of the plots (0.6%, 0.8%, and 1.0%) increased with increasing amplitude, implying a strain hardening of interfaces because more adsorbed LSEP caused molecules to be packed closer together at the interface, hardening the interface and reducing its elastic modulus. These results were in agreement with the trends observed in E-π curves (Figure 6b). The above results showed that the concentration may affect the in-plane interactions between LSEP molecules, thereby reducing the in-plane cohesion of the interface microstructure and affecting the strain softening or hardening behavior of the interface [63]. The results of dilatational rheological measurements showed that the interfacial rheological properties significantly depended on the amplitude. Considering that G′ and G″ exhibited only a slight frequency dependence, the viscoelastic properties were caused by LSEP may be more affected by concentration and amplitude.

## 3. Conclusions

The exopolysaccharide produced by *L. starkeyi* ATCC 58680 was separated, purified, and characterized. Its rheological properties in aqueous solution were also studied. In addition, the adsorption behavior of LSEP at the MCT-water interface was carried out by using an optical contact angle meter. The characterization results revealed a considerable yield (2.85 g/L), a relatively large molecular weight (400 kDa), a semi-crystalline property (60.48% crystallinity), a good thermal stability, and a good water holding capacity (loose and porous cross-linked network structure). In addition, the result of rheological properties showed a higher viscosity and gelling property. Moreover, the results of the optical contact angle meter indicated that the adsorption of LSEP could reduce the interfacial tension well and form an elastic interface layer. This study illustrated that the LSEP possesses a certain potential as a thickener, gelling agent, or emulsifier applied to industrial production.

## 4. Materials and Methods

### 4.1. Materials

The *L. starkeyi* ATCC 58680 was purchased from the American Type Culture Collection (ATCC, Manassas, VA, USA) and revived using yeast mold (YM) medium (Composition: casein peptone 5.0 g/L, malt extract 3.0 g/L, glucose 10.0 g/L, and yeast extract 3.0 g/L). The strain was then cultivated on YM-agar slants (YM medium with 2% agar) at 30 °C for 48 h in an Isotemp Standard Lab Incubator (Tai Hong LRH-250A, China) and stored at 4 °C. The composition of the fermentation medium was (per 1 L of mature coconut water): 90 g of glucose, 0.35 g of KH_2_PO_4_, 0.125 g Na_2_HPO_4_, 1.5 g of (NH)_4_SO_4_, 1.5 g of MgSO_4_, 0.1 g of CaCl_2_, 0.0082 g of FeSO_4_·7H_2_O, 0.01 g of ZnSO_4_·7H_2_O, 0.01 g of MnSO_4_·H_2_O, and 0.01 g of CuSO_4_ [7]. All chemical reagents were purchased from Sigma-Aldrich Chemical Reagent Company. 

### 4.2. Methods

#### 4.2.1. Culture Conditions

The seed was prepared by YM medium with refrigerated cells and incubated at 30 °C for 48 h. Briefly, a loop full of cells from a slant was used to inoculate 125 mL of sterile YM medium in a 500 mL baffled flask. Subsequently, the inoculated medium was placed in a shaker (30 °C, 140 rpm). After cultivation for 48 h, the seed was inoculated (2.5%, *v*/*v*) into the sterilized fermentation medium. The fermentation was performed in a shaker under the conditions of 30 °C and 140 rpm. Sodium hydroxide solution (0.5 M) was used to maintain the pH (5.5) during cultivation [7].

#### 4.2.2. Extraction and Purification of LESP

The LSEP was extracted using the method mentioned in a previous study [34]. After fermentation for 144 h, the culture broth was centrifuged at 12,000× *g* for 10 min to obtain a cell-free supernatant. Subsequently, trichloroacetic acid (4%, w/v) was added to the supernatant and incubated at 4 °C for 6 h to remove the protein. To precipitate the LSEP, pre-cooled 98% ethanol (1:3 volumetric ratio) was mixed and kept at 4 °C for 12–14 h. Furthermore, the LSEP obtained by alcohol precipitation was dialyzed in deionized water using a dialysis bag (Mw cutoff 20,000 Da). After dialysis, the retentate was freeze-dried to obtain LSEP. 

The LSEP solution (100 mg/mL) was subjected to a DEAE-Sepharose Fast Flow column (2.6 × 20 cm, GE Healthcare, Chicago, IL, USA) and eluted with step gradient of sodium chloride (0–0.5 M) at the flow rate of 1 mL/min. The total sugar contents of the collected fractions were measured by the phenol-sulfuric acid method [64]. The selected fractions were pooled, desalted, and further purified using gel-filtration chromatography with a Sephadex-G 200 (1.6 × 50 cm, GE Healthcare, Chicago, IL, USA). The collection of fractions was carried out at a flow rate of 0.5 mL/min using deionized water and the total sugar content in the eluate fractions was tested using the phenol-sulfuric acid method. The purified LSEP was dried and stored for further use.

#### 4.2.3. Physicochemical Characterization of LSEP

##### Monosaccharide Composition Analysis

The monosaccharide composition of LSEP was determined by high-performance liquid chromatography-mass spectrometry (HPLC-MS, Waters UPLC, Waters Xevo TQ-S Micro) after pre-column derivatization [65]. Briefly, LSEP (10.0 mg) was hydrolyzed with 2 mol/L trifluoroacetic acid at 100 °C for 6 h in a sealed tube. Excess acid was removed by co-distillation with methanol four times after the hydrolysis was completed. The dry hydrolysate was dissolved in NaOH (1 mL, 0.3 mol/L) and then added to methanol solution of PMP (400 μL, 0.5 mol/L) at 70 °C for 2 h. Subsequently, HCl solution (400 μL, 0.3 mol/L) was added to the mixture and shaken vigorously. Then an amount of deionized water (1200 μL) and chloroform (1200 μL) was added. The mixture was shaken vigorously, and the chloroform phase was discarded. Finally, the aqueous phase was filtered through 0.45 μm nylon membranes (Westborough, MA, USA), and 2 μL of the resulting solution was injected into the Agilent EC-C_18_ column (2.7 μm, 2.1 mm × 50 mm). The mobile phase A was ammonium acetate buffer (50 mmol/L, pH 8.0) and the mobile phase B was acetonitrile. The flow rate was 0.4 mL/min and the column temperature was 35 °C. The characteristic ion scan (SIR) was used for mass spectrometry scanning. Calculation of the molar ratio of the monosaccharide was carried out on the basis of the peak area of the monosaccharide.

##### Molecular Weight Analysis

The molecular weight (Mw) was determined using gel permeation chromatography (GPC) with refractive index (RI) detector [66] and equipped (1.0 mL/min) with a PolySep-GFC P4000 column (300 × 7.8 mm, Phenomenex). NaNO_3_ (0.2 mol/L) and NaH_2_PO_4_ (0.01 mol/L) were used as the mobile phase. Chromatographic-grade standard dextran with different molecular weights (10, 20, 40, 200, and 500 kDa) was used as the standard substance. The Mw of the LSEP was calculated with the help of a calibration curve (Log MW = 40.053 − 3.93 × V + 0.146 × V^2^ − 0.00191 × V^3^, V means retention time, R^2^ = 0.9966) obtained from dextran standards.

##### Fourier-Transform Infrared Spectroscopy Analysis

The purified LSEP and spectral-grade dry KBr (1:100) were mixed and ground. Subsequently, the mixture was pressed into a round slice. Fourier-transform infrared spectrometer (FT-IR, German Bruker T27) was used to scan and record in the range of 4000–400 cm^−1^ [67].

##### Nuclear Magnetic Resonance Analysis

The purified LSEP (30 mg) was dissolved in D_2_O (99%, 1.0 mL) [68]. A nuclear magnetic resonance (NMR) spectrometer (Bruker Advance 400 MHz) was used to record ^1^H and ^13^C NMR spectra at 25 °C.

##### X-ray Diffraction Analysis

X-ray diffraction (XRD) analysis of LSEP was performed on an X-ray powder diffractometer (Smart Lab, Rigaku, The Woodlands, TX, USA). The scanning was carried out at ranges of 2θ angles (10–70°) with a scan rate of 2°·min^−1^ using Cu-Kα radiation monochromatized with graphite crystals [34].

##### Thermal Analysis

The thermal analysis of LSEP was performed using thermogravimetric analysis (TGA) according to Sran et al. [67]. The TGA of LSEP (10.0 mg) was performed in a nitrogen atmosphere using a thermo-gravimetric analyzer (Pyris 1 TGA, PerkinElmer, Waltham, MA, USA) at a heating rate of 10 °C/min over a temperature range from 30 to 650 °C.

##### Scanning Electron Microscopy Analysis

The purified LSEP was placed on a short aluminum tube with double-sided tape and sprayed with gold. Then, a scanning electron microscope (SEM, Thermoscientific Verios G4 UC) was used to observe and photograph the surface morphology with the resolution at 1000×, 2000×, 5000×, 10,000×, and 20,000× [43].

##### Water Holding Capacity (WHC) and Oil Holding Capacity (OHC) Analysis

The WHC of LSEP was measured according to a reported method [69]. Each sample (0.5 g) was placed into a centrifuge tube to be weighed. Then, an amount of distilled water (50 mL) was added, and the suspensions were held at room temperature for 1 h with stirring for 5 s every 15 min. After 20 min of centrifugation using a refrigerated centrifuge at 5000 *g*, the upper phase was eliminated. Then, the tube was tilted to a 45° angle on a filter paper and drained for 30 min. Each experiment was performed in triplicate. The WHC was recorded according to the following formula: WHC (g/g) = Water absorbed weight (g)/Sample weight (g).

The OHC of LSEP was measured according to the methods reported by Zhao et al. [44]. Briefly, 0.1 g of LSEP was mixed with 10 mL of medium-chain triglycerides (MCTs) in a 25 mL tube. The mixture was stirred with a magnetic stirrer for 1 min and held for 30 min at ambient temperature. The mixture was then centrifuged at 3000× *g* for 25 min. After removing the upper phase, the tube was drained for 30 min on a filter paper and reweighed. Each experiment was performed in triplicate. The OHC value was calculated using the following formula: OHC (g/g) = Oil absorbed weight (g)/Sample weight (g).

#### 4.2.4. Rheological Properties Analysis

In order to investigate the rheological properties of LSEP at different concentrations (2%, 4%, 6%, 8%, and 10%), the LSEP was dissolved in deionized water. Then, the rheological properties of these samples were determined on a rheometer (HAAKE MARS 40, Thermo Fisher, USA) with a 60 mm cone plate (2°). Rotary shear measurements were performed at 25 °C with shear rate from 0.1 to 100 s^−1^. Dynamic strain sweep measurements were performed to determine the linear viscoelastic regimen with a strain range from 0.1% to 100%. The testing conditions were selected to avoid exceeding the linear viscoelastic limits of any samples. The storage modulus (G′) and loss modulus (G″) were measured by oscillation frequency measurements with a strain of 5% and angular frequency ranging from 50 to 0.1 rad/s [50]. The G′ and G″ were fitted by the power law model as follows: G′ = k′ $\omega^{n}{}^{\prime }$, G″ = k″ $\omega^{n}{}^{\prime \prime }$. Where ω is the angular frequency and K′ (K″) and n′ (n″) are the represented intercepts and frequency exponents, respectively [70].

#### 4.2.5. Adsorption Behavior Analysis of LSEP at MCT-Water Interface

##### Interfacial Dynamic Adsorption Analysis

An optical contact angle meter (OCA25, Dataphysics Instruments GmbH, Germany) was used to monitor the interfacial tension at the MCT-water interface (pH 3.5 ± 0.1, 25 ± 1 °C) [71]. LSEP solution (2–10 mg/mL) was contained in the syringe, and the MCT was placed in the cuvette. In order to measure the change of interface tension (γ) caused by the dynamic adsorption process of LSEP at the MCT-water interface, a drop of the LSEP solution (42 μL) was delivered into the cuvette, and the tip of the needle was keep immersed in the oil during the measurement. The dynamic interfacial tension of the MCT-water interface was monitored for 180 min. The image of the drop was continuously fitted and digitized by a charge-coupled device (CCD) camera. The interfacial pressure (π) of MCT-water interface could be calculated as: π = γ_0_ − γ. Where γ_0_ (mN/m) and γ (mN/m) are the interfacial tension of pure oil-deionized water (26.08 mN/m) and LSEP solutions, respectively. The variation in interfacial pressure (π) with adsorption time (t) can be expressed by a modified form of the Ward and Tordai equation [72]: π = 2C_0_·K·T(Dt/3.14)^1/2^, where C_0_ is the concentration in the continuous phase, K is the Boltzmann constant, T is the absolute temperature, and D is the diffusion coefficient. Furthermore, for calculating the rate of adsorption and arrangement of LSEP particles at the interface, the first-order equation was used to fit the data [73]: ln [(π_f_ − π_t_)/(πf − π_0_)] = −k_i_t, where π_f_, π_0_, π_t_, and k_i_ are the interfacial pressure at the final adsorption time, at the initial time, at any time of each stage, and the first-order rate constant, respectively.

##### Interface Dilatation Rheological Analysis

The dilatational viscoelasticity properties of LSEP at the MCT-water interface (pH 3.5 ± 0.1, 25 ± 1 °C) were investigated with an optical contact angle meter (OCA25, Dataphysics Instruments GmbH, Germany). The dilatational viscoelastic modulus (E) was measured as a function of time at constant frequency (0.1 Hz) and amplitude (10%). For each measurement, the cycle frequency was 0.628 rad/s. The value of E was calculated using the equations: σ = σ_0_ sin (ωt + δ), A = A_0_ sin (ω t), and E = dσ/(dA/A), where σ and σ_0_ are the dilatational stresses at initial time and any time, respectively; δ is the phase angle between the stress and strain; A and A_0_ are the interfacial area of the drop at initial time and any time, respectively; and π is the interfacial pressure [74].

In order to investigate the rheological response under various deformations, the amplitude sweeps from 1.5% to 30% deformation were performed at a constant frequency of 0.1 Hz. To perform amplitude sweep, a drop (32 μL) of the LSEP solution was delivered into the cuvette, and the tip of the needle was keep immersed in the oil for 180 min until the quasi-equilibrium conditions were obtained. Then, the shape of the drop was monitored with a video camera, and the γ was calculated from the analysis of a pendent drop according to the Gauss−Laplace equation [75]. The analysis of dilatational amplitude sweeps was performed according to the method developed by van Kempen et al. [76] and Ruhs et al. [61]. The results of amplitude sweeps were presented in the form of Lissajous plots of the surface pressure (Π = γ − γ_0_) versus deformation (δA/A_0_), here δA = A − A_0_; γ and A are the surface tension and area of the deformed interface, and γ_0_ and A_0_ are the surface tension and area of the nondeformed interface.

## Figures and Tables

**Figure 1 gels-07-00156-f001:**
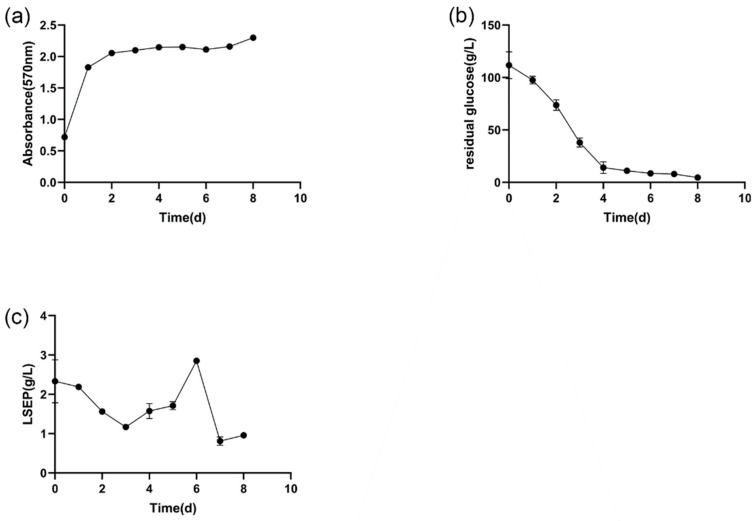
The process of fermentation: the growth of *L. starkeyi* (**a**), the consumption of glucose (**b**), the production of LSEP (**c**).

**Figure 2 gels-07-00156-f002:**
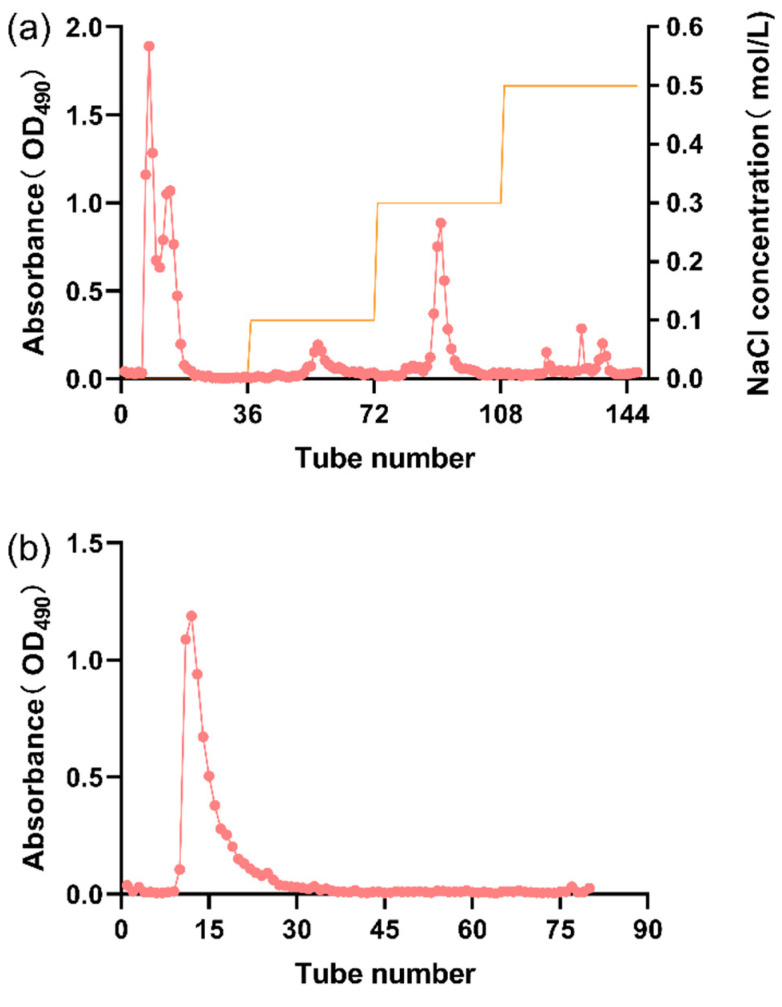
Anion exchange chromatography (**a**) on a DEAE–Sepharose Fast Flow column chromatography and gel permeation chromatography (**b**) on a Sephadex G-200.

**Figure 3 gels-07-00156-f003:**
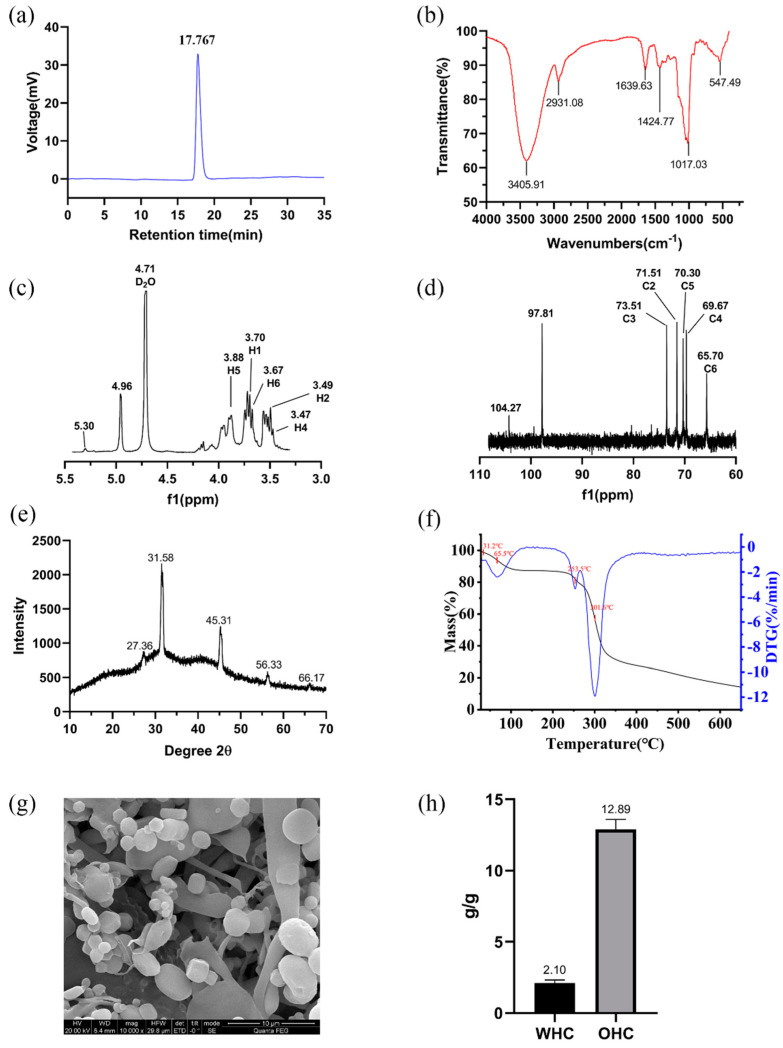
The physicochemical characterization of LSEP: GPC chromatogram of LSEP (**a**), FT-IR spectrum of LSEP (**b**), ^1^H NMR spectrum of LSEP (**c**), ^13^C NMR spectrum of LSEP (**d**), XRD pattern of LSEP (**e**), thermal analysis of LSEP (**f**), SEM image of LSEP magnification 10,000× (**g**), water holding capacity, and oil holding capacity of LSEP (**h**).

**Figure 4 gels-07-00156-f004:**
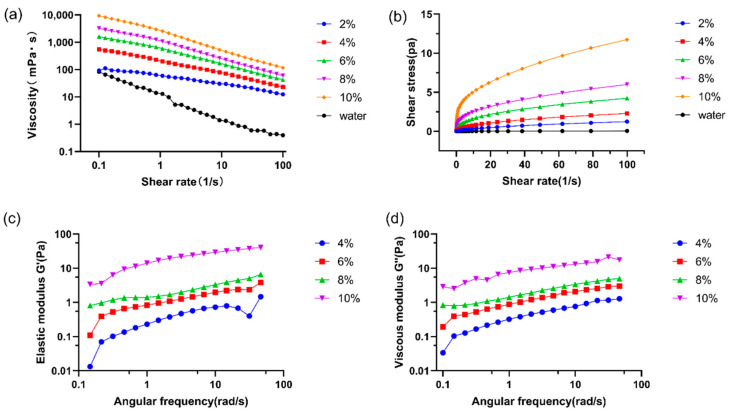
Rheological behaviors of LSEP solutions at 25 °C: Shear-rate-dependent viscosity (**a**) and shear stress (**b**), elastic modulus (G′) (**c**), and viscous modulus (G″) (**d**) as functions of angular frequency (ω).

**Figure 5 gels-07-00156-f005:**
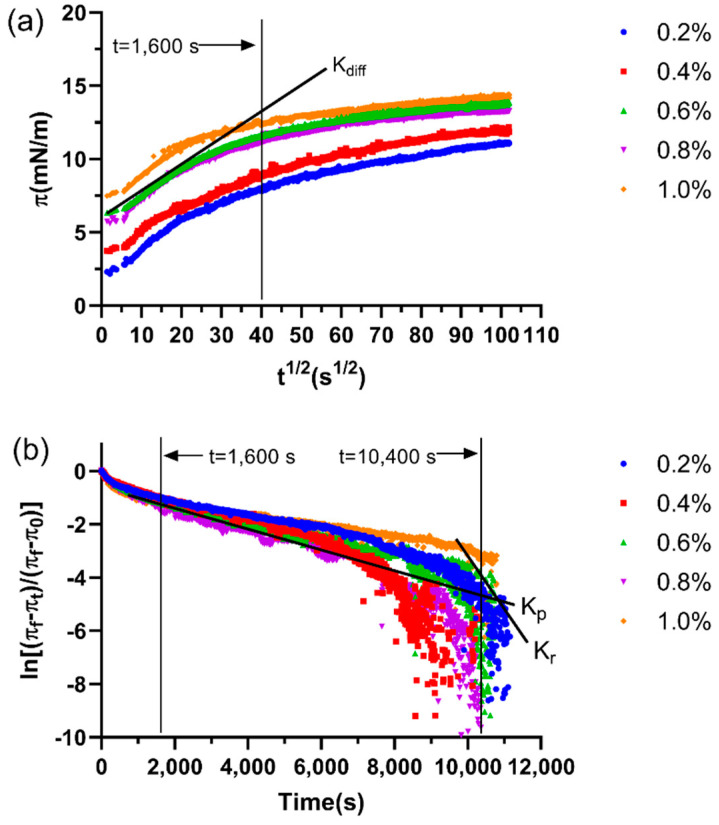
The square root of time (t^1/2^) dependence of interfacial pressure (π) (**a**) and the profile of the molecular penetration and configurational rearrangement steps (**b**) for LSEP at the MCT-water interface. K_diff_ represents the diffusion rate, K_p_ and K_r_ represent the first-order rate constants of penetration and rearrangement, respectively.

**Figure 6 gels-07-00156-f006:**
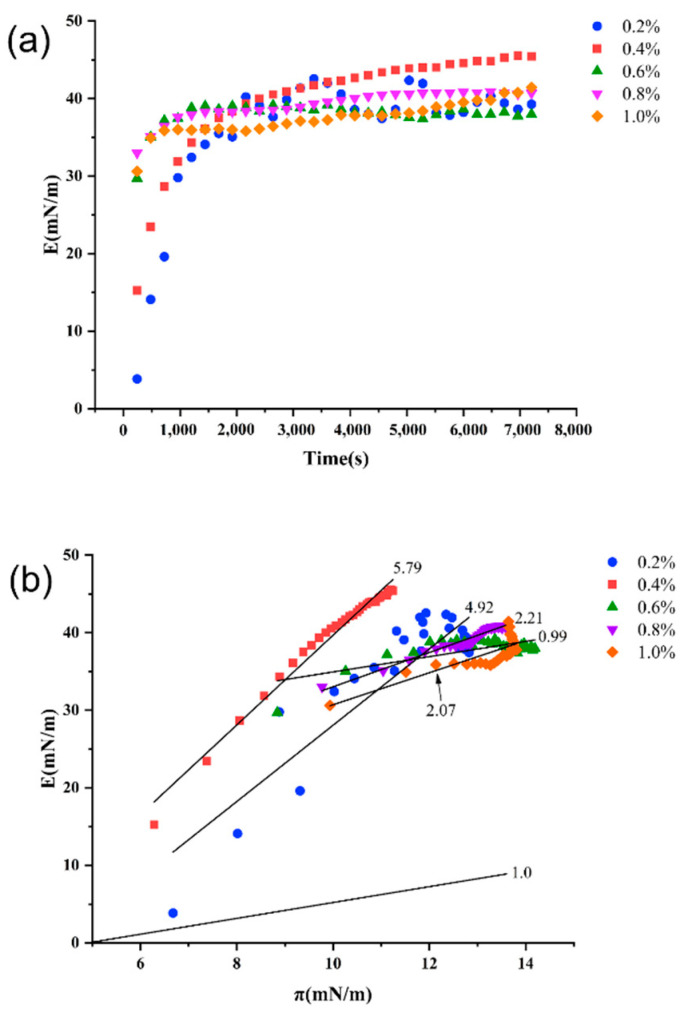
Time-dependent dilatational viscoelasticity modulus (E) (**a**) and the dilatational viscoelasticity modulus (E) as a function of surface pressure (π) (**b**)for LSEP adsorbed layers at the MCT-water interface.

**Figure 7 gels-07-00156-f007:**
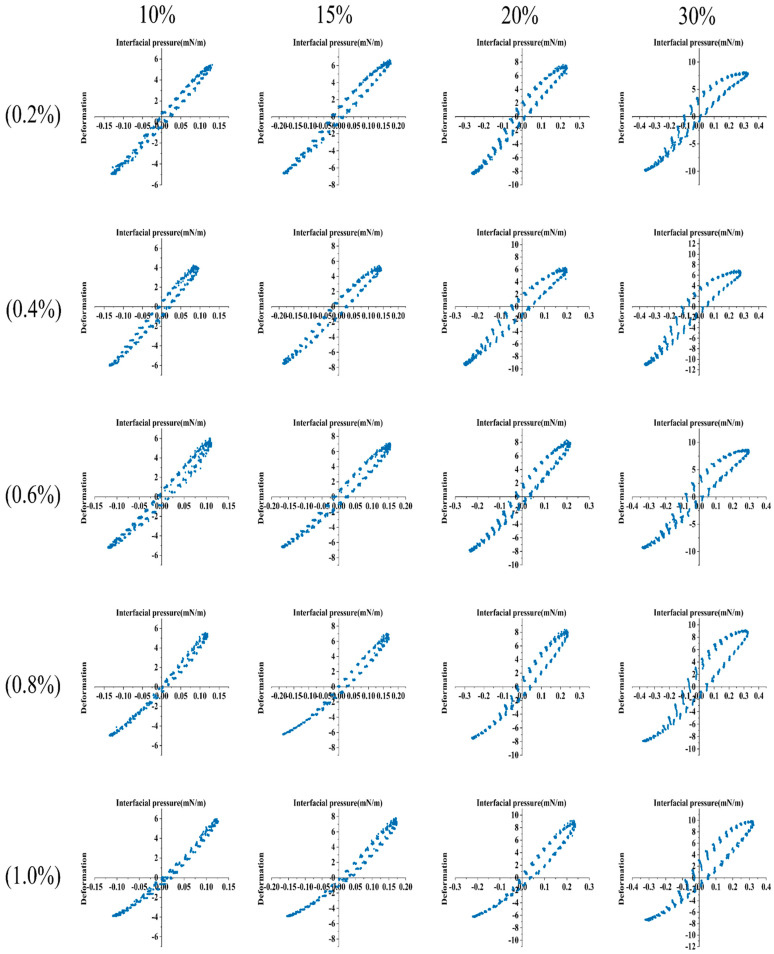
Lissajous plots for different concentrations (0.2–1.0%) of LSEP with various amplitude of deformation (10–30%).

**Table 1 gels-07-00156-t001:** The molecular weight and monosaccharide composition of LSEP.

Sample	Mw (Da)	Monosaccharide Composition (%)			
LSEP	401,813 Da	Mannose	Glucose	Galactose	Arabinose
		8.53%	79.25%	7.15%	5.07%

**Table 2 gels-07-00156-t002:** Parameters of power-law functions describing elastic modulus (G′) and viscous modulus (G″) of the LSEP solutions with different concentrations (4%, 6%, 8%, and 10%).

	k′	n′	R^2^	k″	n″	R^2^
4%	−0.68129 ^d^ ± 0.06137	0.55179 ^a^ ± 0.06959	0.80737	−0.5893 ^d^ ± 0.03231	0.49331 ^a^ ± 0.03663	0.9236
6%	−0.084 ^c^ ± 0.03479	0.44008 ^b^ ± 0.03945	0.89244	−0.09755 ^c^ ± 0.01818	0.40173 ^b^ ± 0.02062	0.962
8%	0.24112 ^b^ ± 0.01251	0.34572 ^d^ ± 0.01419	0.97537	0.17194 ^b^ ± 0.00817	0.33338 ^c^ ± 0.00926	0.98855
10%	1.10715 ^a^ ± 0.02872	0.39933 ^c^ ± 0.03256	0.90931	0.80821 ^a^ ± 0.01637	0.32121 ^d^ ± 0.01856	0.95229

Different letters within the same column are significant (*p* < 0.05, *n* = 3) by Duncan’s multiple range test.

**Table 3 gels-07-00156-t003:** Characteristic parameters, including the apparent diffusion rate (K_diff_), constants of penetration and structural rearrangement at the interface (K_p_ and K_r_), and surface pressure at the initial (π_0_) and the end (π_10,800_) of adsorption for LSEP at different concentrations.

	K_diff_ (mN/m/s^1/2^) (LR)	K_p_ × 10^4^ (LR)	K_r_ × 10^4^ (LR)	π_0_	π_10,800_
0.2%	0.1360 ± 0.00078 (0.9635) ^a^	−3.4781 ± 0.02359 (0.9329) ^b^	−4.3129 ± 0.01831 (0.9576) ^c^	2.32 ± 0.14137 ^e^	11.19 ± 0.01516 ^e^
0.4%	0.1267 ± 0.00042 (0.9818) ^c^	−5.8389 ± 0.00000 (0.85692) ^e^	−7.7266 ± 0.01832 (0.91517) ^e^	3.78 ± 0.06928 ^d^	11.81 ± 0.00985 ^d^
0.6%	0.1367 ± 0.00067 (0.9746) ^a^	−3.6866 ± 0.05457 (0.8553) ^c^	−3.0434 ± 0.01633 (0.8423) ^b^	6.40 ± 0.05828 ^c^	13.89 ± 0.02069 ^c^
0.8%	0.1309 ± 0.00075 (0.9562) ^b^	−5.3614 ± 0.01196 (0.8529) ^d^	−5.1563 ± 0.09415 (0.6121) ^d^	5.77 ± 0.17687 ^b^	13.27 ± 0.00171 ^b^
1.0%	0.1176 ± 0.00114 (0.9213) ^d^	−2.1914 ± 0.01804 (0.8286) ^a^	−2.1689 ± 0.00313 (0.6920) ^a^	7.56 ± 0.08626 ^a^	14.27 ± 0.14834 ^a^

Different letters within the same column are significant (*p* < 0.05, *n* = 3) by Duncan’s multiple range test. LR is an abbreviation for linear regression coefficients.

## Data Availability

Data is contained within the article or Appendix A.

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
