# Peer review of "Physicochemical Characterization of an Exopolysaccharide Produced by Lipomyces sp. and Investigation of Rheological and Interfacial Behavior"

_gels, 2021, doi:10.3390/gels7040156_

Round 1

Reviewer 1 Report

Authors have reported the rheology and interfacial properties of a new polysaccharide material. Some interesting results were presented; however, I have following concerns before suggesting for publication.

  1. My overall feeling of this manuscript is that authors have done a lot of measurements by multiple analytical techniques, but the results were not interpreted and discussed collectively. E.g. the FTIR and NMR just confirm presence of some functional groups, but these signals are interpreted superficially and are completely not surprising. Conclusions are pretty generic such as peaks at some frequencies represent some basic functional groups, and I haven’t seen how authors planned to use these techniques to reveal the characteristic structure of this particular macromolecule. Then what is the purpose of these measurements? Probably they can only prove that this material is a kind of polysaccharide, which may not necessary to prove at all. More importantly, these results are not used to interpret the rheology and interfacial properties in later half of the article, which makes them look redundant. I do suggest to modify this article to make it more integrated.
  2. Section 2.7. “Thermodynamic properties” is not a correct terminology. It refers to state properties of a system such as enthalpy or free energy. Please correct.
  3. Figure 4. It will be good to have a curve of water to view the limit of the instrument. E.g. water may show an apparent shear thinning at low shear rates in rotational rheometers. From figure 4b, I would not call samples at concentration lower than 6% pseudoplastic, the curves are pretty linear.
  4. Figure 4c. I suggest to use logarithm scale in x and y axis, such that more information can be derived rather than simply comparing G’ and G’’. The slope of moduli vs. frequency is also important to reveal the gel-like vs. liquid-like behaviours. Though this is unclear from the manuscript, the test is in SAOS condition I suppose. Authors still need to show that the applied strain/stress is within the linear viscoelastic region. Otherwise the data should be interpreted differently.
  5. Figure 7 is difficult to read. Fonts are too small and words are unclear. Please improve.
  6. I didn’t really get the points about doing nonlinear viscoelasticity measurement at interface, given that the linear viscoelastic data isn’t fully discussed. It gives one parameter dilutional viscoelasticity modulus, whereabout G’ and G’’? Lissajous plots are nice to present, however, they look pretty much identical to me. To reveal the difference if it exists, either move to another frequency or extend to a larger strain. Some derived parameters from LAOS such as small-strain modulus and large-strain modulus may be better and clearer than providing the entire stress-strain loop. The discussion of figure 7 is also superficial and possibly incorrect. Authors mentioned “strain hardening” for multiple times, however, if x-axis is strain and y-axis is stress by convention (sorry they are really unable to read), the plots in figure 7 are all in a shape typical for strain softening samples. This needs to be clarified and corrected if needed.
  7. Similar to the previous comment, if the authors consider necessary to perform LAOS, it is better done in bulk rheology first and then at interface. The rheometer used in this work should be capable enough for doing the LAOS in a similar geometry setup as SAOS.

Reviewer 2 Report

Overall, this is a well-written paper. The authors extracted an exopolysaccharide (LSEP) generated by L. starkeyi using waste matured coconut water as one of major fermentation nutrients. The authors performed a complete characterization to analyze the composition, molecular weight and physicochemical properties, as well as the interfacial behavior when it is applied as a gelling agent. The reviewer has a few comments that would help the authors to further improve the quality of the manuscript.

  1. The objective of applying LSEP as gelling agents was not clear addressed in the introduction. Since the journal focuses on GELS, the authors should add some examples regarding the application of LSEP or related exopolysaccharide as gelling agents, and discuss their advantages.
  2. Does the journal require the manuscript to have the section of Materials and Methods in the last of the text?
  3. What is the mobile phase in HPLC?
  4. Please check if each letter is correctly corresponding to its sub-figure in the caption of Figure 3. Figure 3c/3d may need to add the relevant chemical structure or groups to clearly support the discussion.
  5. What kind of viscosity values were used to describe the rheological behavior? dynamic viscosity, relative viscosity, or absolute viscosity?

Round 2

Reviewer 1 Report

Authors did well improved the article, and have addressed most of my issues. I do not have further concerns of this article. Just few minor ones: 

(1) The inclusion of water in figure 4 shows that water (a Newtonian fluid with low viscosity) is measured as an apparent shear-thinning fluid. This is commonly seen in rotational rheometer and is due to the edge effect (surface tension). This needs to be discussed, as well as how this can possibly influence the observed shear-thinning for other samples. 

(2) Figure 7 is much clear. Some points explained in the response to my previous comments is really good to be included in the article to help to interpret of figure 7. 
